# Assessment of the percentage of full recombinant adeno-associated virus particles in a gene therapy drug using CryoTEM

Mathieu Colomb-Delsuc[1]*, Roman Raim[2], Christian Fiedler[2], Stefan Reuberger[2], Johannes Lengler[2], Rickard Nordström[1], Martin Ryner[1], Ioana Mihaela Folea[1], Barbara Kraus[2], Juan A. Hernandez Bort[2], Ida-Maria Sintorn[1,3]

1 Vironova AB, Stockholm, Sweden, 2 Baxalta Innovations GmbH, A Part of Takeda Companies, Orth an der Donau, Austria, 3 Department of Information Technology, Uppsala University, Uppsala, Sweden

☯ These authors contributed equally to this work.
* mathieu.colomb-delsuc@vironova.com

**Data Availability Statement:** All relevant data are within the paper and its Supporting information files. The source data can be found at the following repository: https://doi.org/10.5061/dryad.8931zcrsp.

## Abstract

In spite of continuous development of gene therapy vectors with thousands of drug candidates in clinical drug trials there are only a small number approved on the market today stressing the need to have characterization methods to assist in the validation of the drug development process. The level of packaging of the vector capsids appears to play a critical role in immunogenicity, hence an objective quantitative method assessing the content of particles containing a genome is an essential quality measurement. As transmission electron microscopy (TEM) allows direct visualization of the particles present in a specimen, it naturally seems as the most intuitive method of choice for characterizing recombinant adeno-associated virus (rAAV) particle packaging. Negative stain TEM (nsTEM) is an established characterization method for analysing the packaging of viral vectors. It has however shown limitations in terms of reliability. To overcome this drawback, we propose an analytical method based on CryoTEM that unambiguously and robustly determines the percentage of filled particles in an rAAV sample. In addition, we show that at a fixed number of vector particles the portion of filled particles correlates well with the potency of the drug. The method has been validated according to the ICH Q2 (R1) guidelines and the components investigated during the validation are presented in this study. The reliability of nsTEM as a method for the assessment of filled particles is also investigated along with a discussion about the origin of the observed variability of this method.

## Introduction

After decades of research and development, gene therapy is nowadays a mature field allowing versatility for the treatment and curation of a broad range of severe and life-threatening diseases [1]. Thanks to this versatility several gene therapy products have been approved by the European Medicines Agency (EMA) and the Food and Drug Administration (FDA) [2, 3] the last few years while thousands of drug candidates are still undergoing clinical trials [4, 5].

**Funding:** Mathieu Colomb-Delsuc, Rickard Nordström, Mihaela Folea, Martin Ryner and Ida-Maria Sintorn are employed by and received salary from Vironova AB. No personnel or department other than the authors played a role in the study design, data collection and analysis, decision to publish, or preparation of the manuscript from Vironova AB. Roman Raim, Christian Fiedler, Stefan Reuberger, Johannes Lengler, Barbara Kraus and Juan Hernandez Bort are employees and received salary from Baxalta Innovations GmbH, a part of Takeda companies. No personnel or department other than the authors played a role in the study design, data collection and analysis, or preparation of the manuscript from Baxalta Innovations GmbH, a part of Takeda companies. A read of the text an approval to publish was performed by the legal department of Baxalta Innovations GmbH, a part of Takeda companies.

**Competing interests:** Authors Mathieu Colomb-Delsuc, Rickard Nordström, Mihaela Folea, Martin Ryner and Ida-Maria Sintorn are all employed by Vironova AB. Vironova AB provides electron microscopy-based imaging and analysis services, as well as image analysis software for this purpose. Authors Roman Raim, Christian Fiedler, Stefan Reuberger, Johannes Lengler, Barbara Kraus and Juan Hernandez Bort are all employees of Baxalta Innovations GmbH, a part of Takeda companies, which are involved in the development of gene therapy products. Employees of Baxalta Innovations GmbH may be owner of stock and/or stock options and may be involved in the filing of patents related to the research presented in this article. The presented work was funded by Baxalta Innovations GmbH.

However, continuous development relating to the efficiency and safety of the strategies is still required [6], implying that there is a demand for objective characterization methodologies.

Recombinant adeno-associated viruses (rAAV) represent one of the most widely used classes of vectors used in gene therapy for the encapsulation and delivery of a genetic sequence of interest [7, 8]. The objective is to; replace a mutated gene with its healthy version, inactivate or repress the expression of mutated genes, edit genes in order to repair defects, or introduce new genes to gain function in cells that help in fighting disease. rAAV vectors have become a prominent vehicle for this purpose as they provide the advantages of a low immune response and are not known to be pathogenic. Clinical trials are currently ongoing for a broad range of diseases using rAAV [5]. For example, Haemophilia B represents an ideal target for which rAAV gene therapy has proven to lead to promising results [9–11], with several drug candidates currently undergoing clinical trials. Small and large animal models can recapitulate this monogenic disease and clinical results have shown a good correlation between the level of blood clotting factor and severity of disease [12–14]. Indeed, rAAV mediated expression of coagulation factor IX (FIX) of one or more percent in plasma as compared to normal already shifts the patient's Haemophilia B status from severe to moderate. Taking into account that a severe haemophilia patient profile corresponds to <1% clotting factor level [15], a clotting factor level increase of 1 to 2 percent in treated patients can be considered therapeutic. Clinical grade rAAV vectors consist of mixtures of empty and filled virions present in variable ratios, varying along the drug manufacturing process. The role of empty particles is not yet fully grasped. Some studies suggest that they can facilitate the gene transfer [16–18] while protocols are developed to reduce their presence as they are commonly considered as contaminants [19], with a potential immunotoxic risk [20, 21] or inhibiting transduction [22–24]. These dichotomies highlight the need for a robust method to quantify the occurrence of each of the particle populations in a gene therapy drug to objectively elucidate their respective roles [25].

For more than two decades now, negative stain transmission electron microscopy (nsTEM) has been used as one of the methods of choice for the assessment of rAAV packaging [26]. This approach allows for direct observation of the particles and a clear discrimination between different particle populations as they appear in the image. More recent studies however have been questioning the reliability and robustness of this method [27–29] and are pointing out inconsistencies of the obtained results [27] due, among others, to the presence of intermediate dense particles [28]. Other analytical methods, such as optical density measurements [30], ion exchange chromatography (IEX) [28], ddPCR [31], qPCR combined with ELISA [32, 33] size exclusion chromatography with multi-angle light scattering detection (SEC-MALS) [34], Analytical Ultracentrifugation (AUC) [35, 36] and more recently charge detection Mass Spectrometry [37, 38] have shown to provide promising results for this assessment. Each of these techniques has its unique strengths and limitations. Some even allow discrimination of sub-populations such as partially filled capsids or capsids with dimeric DNA species. No methodology allowing the separation and identification of these sub-populations has been described using Cryogenic Transmission Electron Microscopy (CryoTEM) yet, and the method introduced in this study only focuses on a binary discrimination of empty and full particle populations. However, in contrast to several of the other aforementioned methods, CryoTEM is considered a platform method, meaning that it is almost not affected by changes in serotype, packaged transgene or composition of the sample matrix. This versatility allows applying the same method without any modifications for different Gene Therapy products which is helpful in the comparative assessment of vector quality and anticipating immunogenic potential by leveraging (pre-)clinical data from Gene Therapy products at more advanced stages of development. As the method is based on a direct visualization of the internal features of particles,

**Table 1. Comparison of orthogonal techniques commonly used to quantitate the relative contents of full and empty AAV capsids.**

|  | cryoTEM | vg/cp ratio | AUC | nsTEM | IEX | SEC-MALS |
|---|---|---|---|---|---|---|
| Throughput[1] | low | high | Low/medium[3] | Medium | high | high |
| Precision[2] | + | - | + | + | + | + |
| Sample volume | < 10 μL | 100–200 μL | > 200 μL | < 10 μL | 100–200 μL | 100–200 μL |
| Sample concentration | > 1E+12 cp/mL | > 1E+08 cp/mL | > 1E+13 cp/mL | > 1E+1111 cp/mL | > 1E+12 cp/mL | > 1E+12 cp/mL |
| Platform capability (serotype/ transgene) | Yes (+/+) | No (-/-) | Yes (+/+) | Yes (+/+) | Partly (-/+) | Yes (+/+) |
| Detection of partially filled sub-population | No | No | Yes | No | No | No |
| cGMP compliance | Challenging[4] | No challenges | Challenging[4] | Challenging[4] | No challenges | No challenges |

[1]Throughput: Low: ≤ 2 samples/day; Medium: 3–10 samples/day; High: ≥ 10 samples/day

[2]%RSD— = > 20%; +/- = 10–20%; + = < 10%

[3] Medium with ≥ 8-place rotor

[4] No or only limited availability of cGMP compliant data analysis software

CryoTEM in combination with image analysis allows an unambiguous discrimination between the particles containing a genome and the particles lacking it, as recently reported for the characterization of rAAV particle packaging [39].

Table 1 provides an overview of methods commonly used for detecting full and empty capsid particles. A recent comparison of these techniques demonstrated good correlation between the listed methods [40]. Within this set of orthogonal methods, AUC is a popular technique due to its capability of discriminating the population of partially filled capsids, a population that is gaining more and more interest in the industry. This species is known to be heterogenous and is subject to further characterization typically trough next generation sequencing or even long read sequencing techniques.

However, this study focuses on analytical methods for the release of clinical grade AAV vectors. Such methods need to fulfil the stringent cGMP requirements, which demands an analysis software that ensures full data integrity and 21CFR part 11 compliance. In this respect AUC is still suffering significant limitations. Since cryoTEM is often applied in research settings, corresponding analysis software typically lack cGMP compliance. In contrast, the cryoTEM method presented here overcomes these limitations and applies a fully validated and 21CFR part 11 compliant software.

In the comparison of orthogonal methods that are commonly used to quantitate the relative content of full and empty capsids in AAV preparations in Table 1, the assessment was based on the experience the authors have gained with these techniques. Some of the parameters evaluated herein might be subjected to evolution as the techniques undergo further developments. Depending on the exact assay set-up the assessment may differ slightly between different laboratories. Due to the lack of an international reference standard for AAV with assigned relative content of full and empty particles, a statement on method accuracy is challenging. In the absence of such a reference standard, an assessment on the alignment and correlation between orthogonal techniques adds valuable information on method reliability.

In the present study, we introduce a complete analytical method based on CryoTEM for quantification of the packaging ratio of a gene therapy drug composed of rAAV8 vectors. The scope of the study is restrained to the rAAV8 serotype, with a focus on proposing a broad set of experiments on a single type of specimen. The CryoTEM images are subjected to image analysis where the rAAV particles are segmented and classified based on their internal density. A CryoTEM correlation study between the derived packaging ratio and the potency of the

drug is performed, validating the importance of the packaging measurement and the presented method. The method is then directly compared with results obtained by nsTEM, where the advantages of CryoTEM in terms of robustness and reliability are highlighted.

This method has been validated according to the principles of Validation of Analytical Procedures, ICH Q2 (R1) [41] and can be used as a quality control in a Good Manufacturing Practice (GMP) workflow. It hence provides an analytical method compatible with established production methodologies and regulatory requirements [27]. Specificity, precision, intermediate precision, linearity, dilutional accuracy, and robustness as defined in ICH Q2 (R1) are assessed. It is the only successfully validated TEM method for the assessment of the percentage of filled particles in an rAAV gene carrier used for gene therapy, hence validating CryoTEM as the method of choice for vector packaging analysis.

## Materials and methods

For this study two types of sample material were investigated. Sample set 1 refers to purified ultracentrifugation fractions with varying degree of full and empty capsids that were directly withdrawn from the production process. For sample set 2 an empty capsid preparation (reference empty) produced by purifying the ultracentrifugation fraction containing primarily empty capsids was mixed at predefined ratio with a preparation containing a high degree of full capsids (reference full) to generate samples with intermediate percentage of full capsids.

### Sample set 1

**AAV production.**   For this study, HEK293 cells adapted to growth in suspension and cultivated in chemically defined serum-free media (FreeStyle™ F17, ThermoFisher, NY, USA), were used to produce rAAV8 vectors. Batch cultures were cultivated in 10 L bioreactors at 37˚C in a humidified atmosphere containing 5% carbon dioxide and with constant stirring at 235 rpm. Transient transfection of HEK293 cells with three plasmids containing Adenovirus 5 Helper genes, Rep2Cap8 and human FIX Padua sequence, respectively, was carried out with Polyethylenimine (Merck KGaA, Darmstadt, Germany) following the supplier transfection protocol. Five days after transfection, the fermentation broth containing rAAV8 particles was separated from cell debris with a clarification step, followed by ultrafiltration and diafiltration step for product concentration and buffer exchange while also reducing impurities. AEX chromatography was used to continue reducing negatively charged impurities followed by a second ultrafiltration and diafiltration step to further concentrate the product and remove protein impurities. The concentrated and diafiltrated intermediate containing rAAV8 was subsequently processed with an ultracentrifugation step according to a Takeda proprietary buffer and protocol, for separation of full and empty particles.

**Sample preparation.**   Applying Takeda's proprietary ultracentrifugation process (WO 2018/128688 A1) full rAAV8 particles were isolated from the empty particles and collected separately: The ultracentrifugation step was performed using a core filled with 50% product + 50% TBS/sucrose buffer) for 6 hours at target rotation speed of 35000 rpm. The AAV8 particles contained in the starting material move into the TBS/sucrose buffer gradient until they reach a point at which their density matches the density of the surrounding gradient. As full and empty capsids differ in their density, this feature is used for the separation of full and empty virus capsids.

For sample set 1, the gradient after centrifugation was subdivided in several fractions with decreasing density. Each fraction was further purified by Ion exchange chromatography and subsequently analysed by cryoTEM, qPCR, and ELISA. Table 2 allows for direct comparison of full and empty capsid analysis through cryoTEM and calculation of the ratio of vector

**Table 2. Sample set 1.**

| Sample ID | cryoTEM [%Full capsids] | FIX-qPCR [vg/fraction] | AAV8 ELISA [cp/fraction] | qPCR/ ELISA ratio [%Full capsids] |
|:---:|:---:|:---:|:---:|:---:|
| **S1.1** | 2 | 1.79E+14 | 3.49E+15 | 5 |
| **S1.2** | 5 | 1.73E+14 | 3.75E+15 | 5 |
| **S1.3** | 10 | 1.28E+14 | 1.35E+15 | 9 |
| **S1.4** | 11 | 1.42E+14 | 8.50E+14 | 17 |
| **S1.5** | 18 | 1.16E+14 | 5.25E+14 | 22 |
| **S1.6** | 26 | 7.24E+13 | 2.89E+14 | 25 |
| **S1.7** | 43 | 4.25E+13 | 6.19E+13 | 69 |
| **S1.8** | 59 | 5.40E+13 | 6.47E+13 | 84 |
| **S1.9** | 70 | 7.81E+13 | 1.06E+14 | 74 |
| **S1.10** | 79 | 4.59E+14 | 3.81E+14 | 120 |

genomes (vg) as determined by qPCR and capsid particles (cp) as quantified by ELISA. However, while both methods show the same trend there is not a one-to-one correlation between the methods. This meets the expectations as both methods rely on different technologies for determining the percentage of full and empty capsids.

## Sample set 2

For this study, HEK293 cells adapted to growth in suspension and cultivated in chemically defined serum-free media (FreeStyle™ F17, ThermoFisher, NY, USA), were used to produce rAAV8 vectors. Batch cultures were cultivated in 10L bioreactors at 37°C in a humidified atmosphere containing 5% carbon dioxide and with constant stirring at 235 rpm. Transient transfection of HEK293 cells with three plasmids containing Adenovirus 5 Helper genes, Rep2Cap8 and human FIX Padua sequence, respectively, was carried out with Polyethylenimine (Merck KGaA, Darmstadt, Germany) following the supplier transfection protocol. Five days after transfection, the fermentation broth containing rAAV8 particles was separated from cell debris with a clarification step, followed by ultrafiltration and diafiltration step for product concentration and buffer exchange while also reducing impurities. AEX chromatography was used to continue reducing negatively charged impurities followed by a second ultrafiltration and diafiltration step to further concentrate the product and remove protein impurities. The concentrated and diafiltrated intermediate containing rAAV8 particles was subsequently processed with an ultracentrifugation step according to a Takeda proprietary buffer and protocol, for separation of full and empty particles. While for samples **S2.1** and **S2.6** ultracentrifugation fractions with high degree of full particles were collected, for sample **S2.5** fractions with mainly empty capsids were pooled for further processing.

Subsequently, product fractions (**S2.1**, **S2.5**, **S2.6**) were further purified by Ion exchange chromatography and IEX Eluates for samples **S2.1** and **S2.5** were nanofiltrated. The resulting starting materials for the spiking (**S2.1** "reference full" and **S2.5** "reference empty") were thoroughly analysed. The mixed samples with intermediate degrees of full particles (**S2.2**—**S2.4**) were prepared based on the AAV ELISA titres as well as on an initial CryoTEM result. The "reference empty" material was prediluted to adjust for differences in particle concentration followed by the mixing with the "reference full" sample to yield the theoretical degree of full capsids with a nominal value of 40%, 60% and 70% full particles. The list of specimens analysed in this study are summarised in Table 3.

**Table 3. Samples used for the specificity, linearity and comparative studies.** Theoretical value calculations are described in section 4 of the Supplementary information.

| Sample ID | Theoretical percentage of full particles | In-vitro BP [rel. BPU] | In-vivo BP [IU/mL] | Imaging method | Type of analysis |
|---|---|---|---|---|---|
| S2.1 | 79.32% (reference full) | 1.61 | 7.70 | CryoTEM | Specificity / Linearity |
| S2.2 | 71.30% | 1.33 | 6.63 | CryoTEM | Specificity / Linearity |
| S2.3 | 61.28% | 1.37 | 6.07 | CryoTEM | Specificity / Linearity |
| S2.4 | 41.24% | 0.74 | 3.15 | CryoTEM | Specificity / Linearity |
| S2.5 | 1.17% (reference empty) | 0.00 | 0.20 | CryoTEM / nsTEM | Specificity / Linearity / Method comparison |
| S2.6 | n/a; rich in full particles | n.d. | n.d. | nsTEM | Method comparison |

Specimen **S2.1** was used as the "full reference" specimen, and specimen **S2.5** as the "empty reference." Specimens **S2.2**, **S2.3** and **S2.4** were prepared by mixing specimen **S2.1** "reference full" and **S2.5** "reference empty." Specimens **S2.5** and **S2.6** were rich in empty and full particles, respectively, and used for directly comparing CryoTEM and nsTEM.

## AAV8 ELISA

The commercially available enzyme-linked immunosorbent assay (ELISA; Progen AAV8 titration ELISA kit, cat. No. PRAAV8) uses a monoclonal antibody (ADK8) specific for a conformational epitope on assembled AAV8 capsids. This plate-immobilized antibody captures AAV-8 particles from the specimen. Captured particles are then detected by the binding of biotinylated anti-AAV8 ADK8 since the epitope targeted is repeatedly expressed on the assembled AAV8 capsid. Streptavidin peroxidase and a peroxidase substrate is then used for measuring bound anti-AAV8 and thus the concentration of AAV8 capsid. The color reaction was measured photometrically at 450 nm. The kit contains an AAV2/8 particle preparation as calibration standard with a labelled AAV8 particle concentration. However, an internal reference standard was used for quantification of capsid particles. This standard was assigned a µg/mL value based on densitometric analysis of a Coomassie stained SDS-PAGE. The assigned µg/mL value was then correlated against the ATCC recombinant Adeno-associated virus 8 (VR-1816) standard, which led to a conversion factor of 8.5E+13 cp/mg.

## FIX-qPCR

Samples were treated with DNAse (NEB) to remove extraneous ITR target sequences. After treatment with Proteinase K (NEB) the AAV genome is released from the capsid. Subsequent restriction enzyme digest with BssHII (NEB) was performed to resolve AAV ITR T-shape structures. For quantification, a Taqman-based method with FIX-specific primers and probe was used.

Forward primer: 5'-CCG GTA CGT GAA CTG GAT CAA-3' Thermo Fisher Scientific

Reverse primer: 5'-CAG CGA GCT CTA GGC ATG CT-3' Thermo Fisher Scientific

Probe 5'-6FAM-AAA CCA AGC TGA CCT GAT MGB-3' Thermo Fisher Scientific

Plasmid encoding for FIX transgene as well as regulatory elements was linearized by ScaI (NEB) restriction digest, separated on Agarose Gel, and the band specific for full length vector genome was purified from gel (QIAquick Gel Extraction Kit) to serves as reference standard.

## Biopotency assays

For the determination of *in vitro* biopotency, HepG2 cells were infected in duplicates with AAV8 vector carrying FIX transgene as described previously [42]. Subsequently, the

chromogenic activity of FIX in the supernatant was measured using the Rox Factor IX kit (Rossix AB, Moelndal, Sweden). The results are given as biopotency unit (BPU) representing the relative FIX activity at a dose of 3.27 x $10^3$ capsid particles (cp) per cell. In the course of the *in vivo* biopotency assay, 2.47 x $10^{11}$ cp/kg of FIX encoding AAV8 particles were infused intravenously into seven hFIX-knockout mice (B6;129P2-F9$^{tm1Dws}$) per group [43] that were bred by Charles River GmbH (Sulzfeld, Germany) and kept as described [44]. Human FIX activity in mouse citrate plasma drawn at day 14 was tested by a one-stage activated partial thromboplastin time (APTT) assay using human FIX-deficient plasma as substrate as already described [43]. Results refer to a human plasma standard which was calibrated against an international standard.

## TEM specimen preparation and imaging

**CryoTEM.** 400 mesh copper grids coated with a carbon film, overlaid with a Formvar® film (TedPella, Inc., CA, USA) were hydrophilized using a glow discharger (Pelco EasiGlow™, TedPella Inc., CA, USA). A glow discharged grid was mounted on a vitrification robot (Vitrobot™ Mark II, FEI, OR, USA). 3 µL of sample were placed onto the grid in the specimen chamber of the vitrification robot, under temperature and humidity-controlled conditions (16˚C, 99% relative humidity (rH)). After an adsorption time of *ca.* 10 sec, the grid was blotted-off and subsequently plunge-frozen in liquid ethane. The specimen was then stored in liquid nitrogen until insertion in the microscope.

The frozen specimen was mounted onto a Gatan 626 single tilt cryo holder (Gatan Inc., CA, USA) under cryogenic conditions and subsequently inserted in a CM200 electron microscope (Phillips N.V., Eindhoven, The Netherlands) equipped with a Field Emission Gun operating at 200 kV. The grid was visually assessed and several areas containing thin amorphous ice, suitable for imaging, were identified. Low-dose images were acquired in areas representative of the sample, i.e. areas containing AAV particles with minimal presence of ice contaminant and a homogenous spreading of particles in thin amorphous ice. The images were acquired at a resolution of 2048 x 2048 pixels using a TVIPS F224HD camera (Tietz Video and Image Processing Systems GmbH, Gauting, Germany).

**Negative stain TEM.** 400 mesh copper grids coated with a carbon film, overlayed with a Formvar® film (TedPella, Inc., CA, USA) were hydrophilized using a glow discharger (Pelco EasiGlow™, TedPella Inc., CA, USA). A glow discharged grid was mounted on tweezers and 3 µL of sample were placed onto the grid. After an adsorption time of *ca.* 10 sec, the excess of liquid present on grid was blotted-off using Grade 1 filter paper pre-wetted with 12 µL of water and immediately washed with 3 µL of distilled water. Excessive liquid was then blotted-off after *ca.* 10 sec and 3 µL of Uranyl acetate 2% (Electron Microscopy Sciences, PA, USA) was immediately added to the grid and blotted-off after *ca.* 10 sec. The grid was then inserted in a Tecnai G$^2$ Spirit BioTwin electron microscope (FEI, OR, USA) equipped with a tungsten filament operating at 100 kV. The grid was visually screened and areas suitable for imaging were identified. Representative images were subsequently acquired at a resolution of 2048 x 2048 pixels using a Veleta CCD camera (Olympus, Tokyo, Japan).

## Image analysis

**Particle detection.** The acquired images were analysed using VAS (Vironova Analyzer Software, Vironova AB, Stockholm, Sweden), a 21CFR part 11 compliant validated software. The particle detection settings used are shown in S4 and S5 Tables. Briefly, potential rAAV particles within a size range of 18–28 nm and 25–35 nm for the CryoTEM and nsTEM images respectively, were detected by local ellipse detection in the gradient magnitude image. The

difference in the size range used between the two analysis methods can be explained by hypotheses such as the presence of stain in nsTEM surrounding the particle, thus increasing the apparent size, by the changes in osmotic effects due to the stain, inducing a swelling of the particles or by a flattening of the AAV particles on the support in nsTEM upon embedding in the stain. A minimum of 3 images per sample were processed until at least 1500 particles were detected. Manual curation and verification of the images and detected particles were performed to remove falsely detected particles and add non-detected particles.

**Particle classification.** Principal component analysis (PCA) was performed on the particles' radial density profiles (RDPs), i.e. the mean intensity at each radial distance from a particle's centre. In a 2-D scatter plot of the two principal components, two separable clusters are observed, one corresponding to full particles, the other one corresponding to empty particles. A particle class ("Full" or "Empty") was attributed to each of the clusters by selecting manually the particles of one cluster on the displayed plot and assigning them the corresponding class. Particles for which the classification was not unambiguous were attributed the class "Uncertain". The corresponding data was extracted from the plot and used for the analyses. Intermediate results from the different steps of the image analysis workflow are shown in section 5 of the Supplementary information.

## Results

### Characterization of the particle populations present in the specimen

Recombinant AAV vectors of serotype 8 (rAAV8) containing a gene coding for FIX were analysed by CryoTEM. Visual analysis of the rAAV preparation **S2.1** images reveal that two main populations of particles can be observed in the specimen. A first population exhibits a distinct outer layer with a minute internal density, appearing as dark circles in the images, while a second population of particles displays a uniform inner density with no clear distinction between the inner core and the edges of the particles, appearing in the images as homogenous dark disks. This second population supposedly represents full particles, i.e. particles filled with material, hence the higher internal density, while the particles appearing as circles are empty particles (Fig 1a). Note that in most of the samples analysed, a small proportion of the observed particles appear as incomplete or improperly assembled (Fig 1b). Particles sharing features of both classes, which could represent intermediately filled particles, were seldom observed in

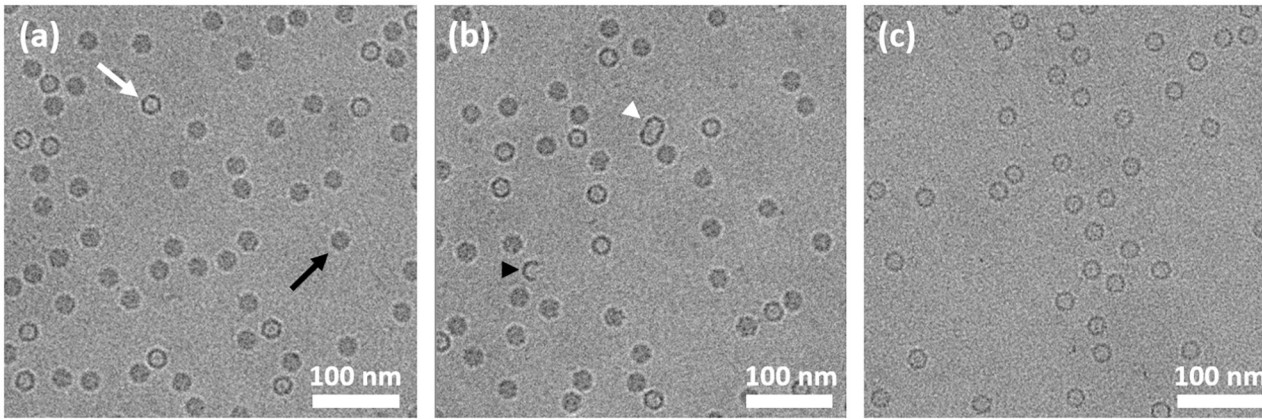

**Fig 1. CryoTEM images of specimen S2.1.** (a) Representative CryoTEM image of specimen **S2.1**, containing filled AAV particles (black arrow) and empty AAV particles (white arrow). (b) CryoTEM image of **S2.1** in which incomplete particles (black arrowhead) as well as improperly assembled particles (white arrowhead) are observed. (c) Representative CryoTEM image of AAV specimen **S2.5**, reference empty. Scale bar represents 100 nm.

this study, as shown with the high confidence interval displayed in the image analysis. As the characterization of such particles is not unambiguous, they are however classified as "uncertain" in the image analyses.

Representative sets of images were acquired for each sample and subjected to image analysis to evaluate the percentage of full particles according to the above definitions present in the specimen. A population of at least 1500 particles was included in each dataset, this value having been evaluated to provide robust statistical relevance (See supplementary section 7). For the experiments in which replicates were used, the mean value of the analyses is reported for each population. Note that the particles identified as broken or incomplete, representing a neglectable amount of the observed particles were not included in the image analysis. Indeed, adding such particles to the analysis would induce a bias towards the empty population as some particles putatively fully can lose their cargo upon breakage while empty particles remain empty upon breakage. Moreover, the discrimination between broken particles and other specimen debris might lead to erroneous interpretation. Including such particles in the analysis might be relevant for the evaluation of the stability of AAV particles or the resistance to stress between the particle types, but this is not within the scope of this study, hence the exclusion of such particles. For the same reasons, particles appearing as broken AAV doublets or triplets were not included in the analysis.

## Correlation of CryoTEM results to biopotency assays and qPCR/ELISA quantification

First, we investigated the correlation of the biopotency and the degree of full rAAV capsid particles as determined by CryoTEM. For this purpose, samples with intermediate ratios of full and empty capsids were prepared by mixing a rAAV preparation containing mostly empty capsids with one containing predominantly full capsids (S2 samples). *In vitro* biopotency results refer to the infection of 3.27 x $10^3$ cp/cell, and *in vivo* dosing was performed at a dose of 2.47 x$10^{11}$ cp/kg. Both the *in vitro* and *in vivo* experiments demonstrated that the biopotency of the drug per capsid particle increases with the percentage of full particles. The results, presented in Fig 2, suggest a clear correlation between biopotency of the drug and the percentage

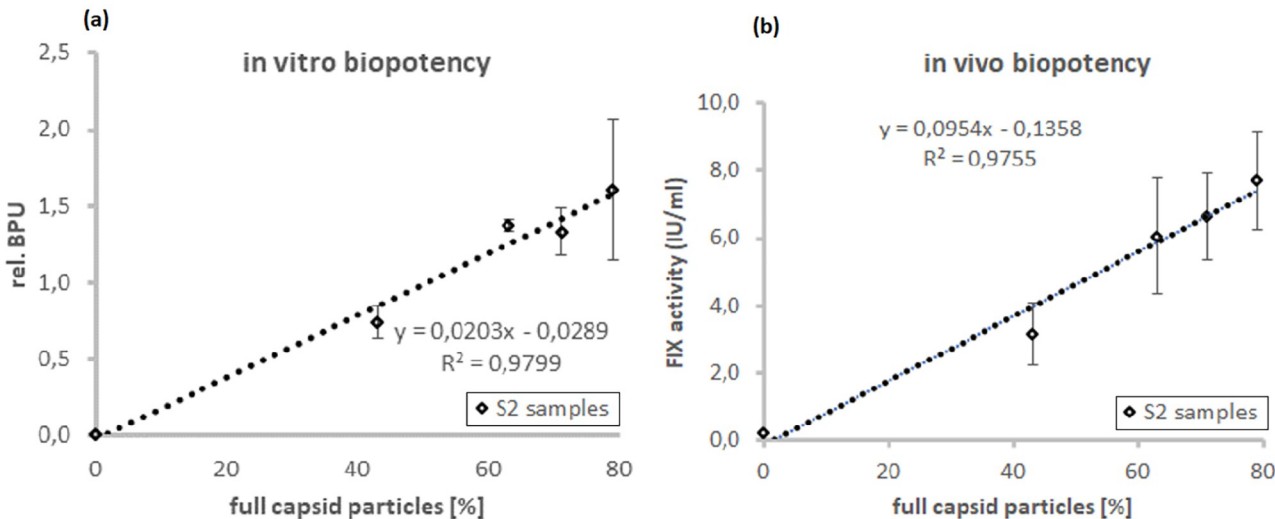

**Fig 2. Correlation between CryoTEM and biopotency.** Correlation between the in-vitro biopotency (a) and in-vivo biopotency (b) of the drug and the percentage of full particles as detected by CryoTEM for a set of specimens with varying ratios of full particles.

of full particles observed by CryoTEM. This observation underlines the reliability of the results generated by cryoTEM and emphasizes the importance of controlling the ratio of filled particles for reaching consistent biopotency at consistent level of empty particles.

Furthermore, good correlation of CryoTEM data with the orthogonal determination of full and empty capsids by calculation of the ratio of vector genomes and total particles was demonstrated and is shown in Fig 3. However, it should be noted that the latter approach bears the risk of inaccurate (e.g. significant over- or underestimation of the full and empty ratio) data due to the nature of the underlying methods (e.g. dependency of PCR based methods on location of amplification target or dependency on reference standards). This potential inaccuracy of qPCR and ELISA based methods is a possible explanation to that the slope of the linear regression as depicted in Fig 3 is > 1.

This indicates that the vg/cp ratio finds higher percentage of full capsids compared to cryoTEM and that this is (more pronounced for preparations with higher degree of full capsids. Another reason for this observation may be the presence of a population of AAV vectors that are detected in PCR based techniques but not in cryoTEM. Such a population may come from AAVs packaged with incomplete transgenes of relatively small size and therefore detected as empty by cryoTEM but still carrying the PCR target sequence.

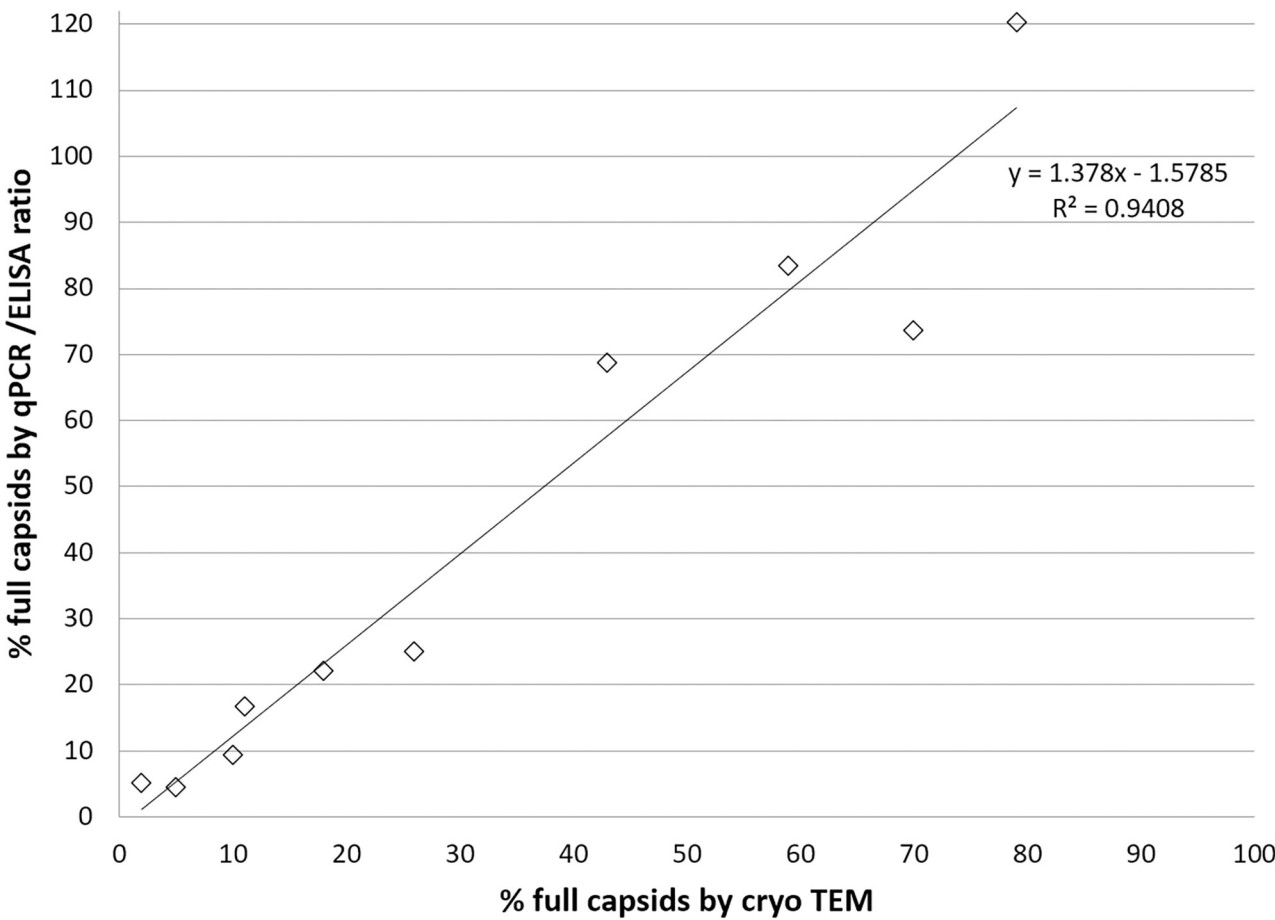

**Fig 3. Correlation between CryoTEM and qPCR/ELISA.** Sample Set 2 with increasing degrees of full capsids were purified from ultracentrifugation fractions. Vector genomes per capsid particle titres were calculated from qPCR and AAV8 capsid particle ELISA results and correlated to the CryoTEM data.

### Repeatability and linearity

The reliability of the CryoTEM method was evaluated by performing a repeatability study. Six grids of the same sample were consecutively prepared and imaged, aiming to assess the analytical method variance under the same operating conditions over a short interval of time. The results show a relative standard deviation of 0.6% (see S2 Table), suggesting a robust repeatability of the method with a low variance. These results are comparable to the values for AUC, where a standard deviation of 0.7% corresponding to a calculated relative standard deviation of 1.8% is reported [35].

The linearity of the method within a range was then evaluated to assess the ability to obtain test results which are directly proportional to the percentage of filled particles in the sample by CryoTEM. Two AAV preparations, used as reference were mixed to obtain a set of specimens with intermediate contents of full particles. AAV preparation **S2.1** was used as the reference full specimen, packed with the best achievable ratio of full particles, containing 79.3 ± 0.8% of full particles and referred to as "reference full" while preparation **S2.5**, containing 1.17 ± 0.37% of full particles was used as the reference empty specimen, containing the lowest achievable ratio of full particles and referred to as "reference empty" (Fig 1, S1 Appendix). Since Cryo-TEM is the first validated method as per ICH Q2 (R1) guidelines, no orthogonal method was available for accuracy assessment, hence the linearity experiments are based on dilution series and the accuracy limited by dilutional accuracy.

A total of 6 grids were prepared for CryoTEM and imaged for each of the 5 samples (**S2.1**—**S2.5**) investigated. Fig 4 shows the correlation between the data obtained experimentally and the estimated theoretical values (see section 4 in Supplementary information). The method appears to display a robust linearity, with the two particle populations being detected to a same extent regardless of their respective concentration in solution, and is therefore deemed reliable within the range evaluated, i.e., at least up to percentages of 79% of full particles present in the

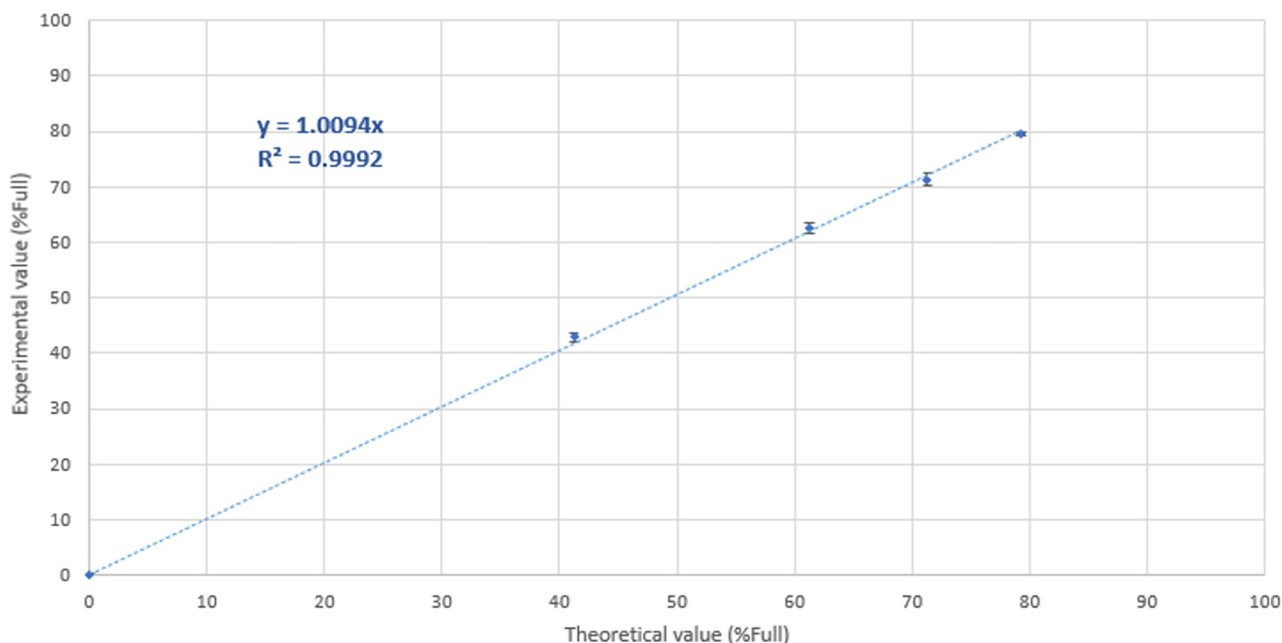

$$y = 1.0094x$$
$$R^2 = 0.9992$$

**Fig 4. Relation between the theoretical filled values and the obtained experimental data.** Each data point from the plot corresponds to the mean value of twelve repeats (*cf.* S3 Table), with the error bars corresponding to the standard deviation of each population.

specimen. As the image analysis implies a binary discrimination between two well identified populations, with no overlap of the 99% confidence interval from the principal component analysis of the radial density profile the of the two classes, one can postulate that the linearity of this method could be extrapolated to be valid up to specimen containing exclusively full particles. This assumption, however, remains to be confirmed experimentally.

## Comparison between CryoTEM and nsTEM

Negative stain transmission electron microscopy (nsTEM) has been routinely used for the assessment of the ratio of filled AAV particles for more than two decades [26]. Unlike in CryoTEM where the specimen is quickly plunge-frozen into a cryogen, allowing its conservation and observation in a near native hydrated state, the preparation in nsTEM involves a staining step, where a salt containing electron rich elements is added, acting both as a contrasting agent and as an embedding agent. The preparation involves drastic local osmotic and pH changes which might affect the particles integrity, and the conservation of the prepared specimen is then done at room temperature which might result in dehydration in areas of the support where the embedding agent was not efficiently spread. As this method is routinely used for the quantification of the percentage of full particles, it was therefore evaluated in the early stages of the drug development of the rAAV8 particles described in this study. When assessing results obtained by nsTEM and CryoTEM, discrepancies were observed. The results presented in Fig 5 suggest that the population of particles appearing as bright with a smooth surface, typically referred to as "full" [22], are over-represented in the nsTEM images, constituting the main population in all samples analysed, with values of 98% for compound **S2.6** and 100% for compound **S2.5**, when CryoTEM results show that specimen **S2.1** contains 83% "full" particles

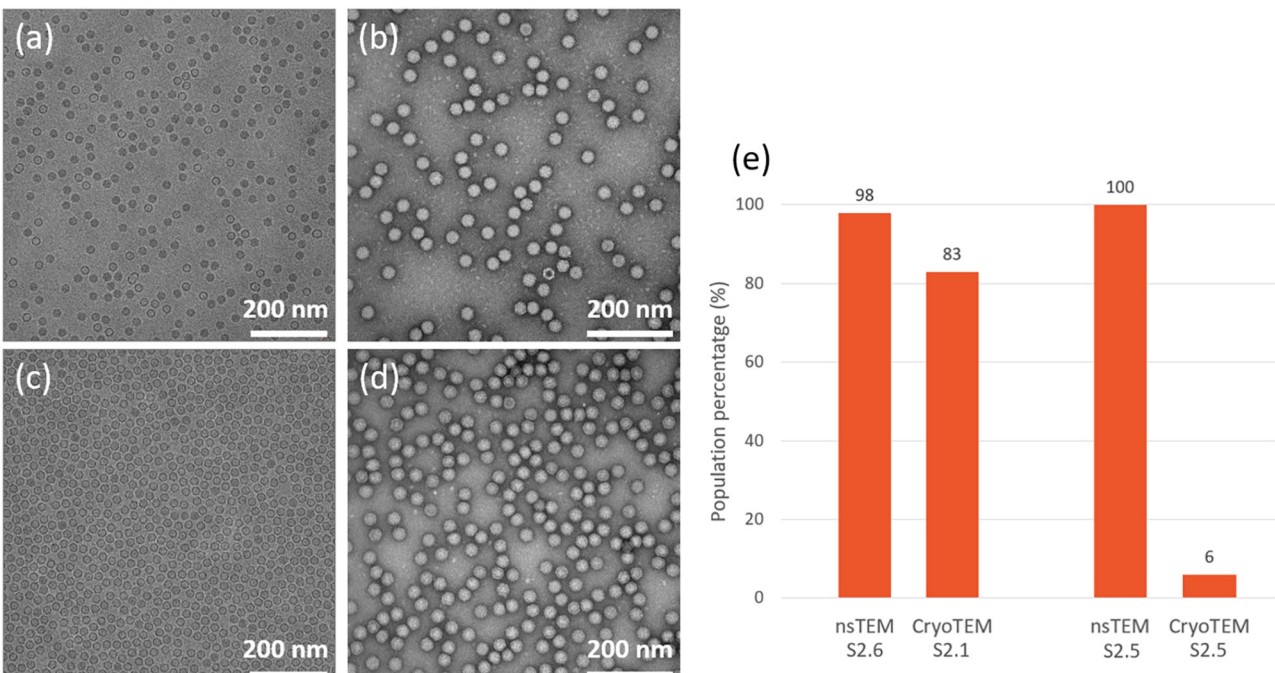

**Fig 5. Comparison of specimens in CryoTEM and nsTEM.** (a) CryoTEM image of specimen **S2.1**, rich in AAV particles carrying a genome. (b) nsTEM image of specimen **S2.6**. (c) CryoTEM image of specimen **S2.5**, rich in empty particles. (d) nsTEM image of specimen **S2.5**. Scale bar represents 200 nm. (e) Comparison on the assessment of the percentage of full particles between nsTEM and CryoTEM. The population of full particles in specimen **S.2.5**, rich in empty particles, is overrepresented in nsTEM. Scale bar represents 200 nm.

while compound **S2.5** contains mainly empty particles, with only 6% of the particles identified as full.

## Discussions

The CryoTEM results presented herein relate to characterizing particle populations, linearity and repeatability of full particles in a rAAV preparation. The proposed CryoTEM method has been validated according to the principles of Validation of Analytical Procedures, ICH Q2 (R1)) as used in a GMP workflow [41], for the assessment of the percentage of full particles in a rAAV8 preparation. As no other validated methods were available at the time of the study, the assessment of the method accuracy was challenging. However, this study can now facilitate the validation of any new methodology, as it will be possible to use CryoTEM analysis as an orthogonal method.

Observations of discrepancy between nsTEM and other orthogonal methods e.g. AUC have been reported [29], without further insights on the origin of these disparities. One hypothesis to explain the differences observed between nsTEM and CryoTEM in this study resides in the way the particles are identified and defined. In nsTEM, the particles that display an inner density either contain stain or could be collapsed particles with stain deposited on top of them, indicating that they do not contain a gene and are thereof indeed empty, as per the commonly reported definitions. On the other hand, particles with an even density do not contain any stain and are not collapsed, indicating that they are intact, are commonly referred to as full. However, their apparent intactness does not provide information about their content, for a particle could be intact and packed with a gene (i.e. full), or still be intact but empty, but with the current definitions, both would be classified as full. On the contrary, in CryoTEM, no stain is added, and the specimen is preserved in a hydrated native state [45], enabling a direct correlation of the inner density of the particles to their packaging level. An AAV particle containing a gene will be electronically denser than an empty one, hence appearing darker on the image. The proposed hypothesis is therefore that in the case of the sample analysed here by nsTEM, the evaluation of the content of full particles might be biased by an incorrect classification of the particles, resulting in an over-expression of the population of full rAAV particles, leading to significant divergences between CryoTEM and nsTEM when the particles are well embedded in stain as observed in Fig 5. For these reasons, nsTEM was not chosen as a method of choice in the case of the specimen studied herein.

The results presented in this study demonstrate the relevance of CryoTEM as a robust and reliable method for the assessment of the percentage of filled particles in a recombinant rAAV8 gene delivery drug product. The repeatability, linearity and robustness of the method demonstrates the suitability of this method as a method of choice that can be used during the different phases of the drug development process as well as validation and quality control in the production phase. Although the scope of the study was restrained to the rAAV8 serotype, available data from CryoTEM analyses of other serotypes (e.g. AAV9) have indicated that findings within the scope of this study can likely be leveraged to other serotypes as well (See S2 Fig). In the presented comparison, nsTEM proved to be a poor technique for the assessment of the content of filled particles for this sample type. Potency data are in line with the results obtained by CryoTEM, reinforcing the reliability of the method. Further studies to broaden the scope of the method to other rAAV serotypes and other types of carriers could prove valuable. More developments could also be performed using CryoTEM, exploring new approaches to allow the discrimination of several levels of particle packaging, including partially filled rAAV particles, both from a preparation and an image analysis perspective. Such advances would benefit the methods in place for the characterization of rAAV drug products.

## Supporting information

**S1 Table. Specificity study data.**
(PDF)

**S2 Table. Repeatability study data.**
(PDF)

**S3 Table. Linearity study data.**
(PDF)

**S4 Table. VAS detection settings for the image analysis of CryoTEM AAV particles.**
(PDF)

**S5 Table. VAS detection settings for the image analysis of nsTEM AAV particles.**
(PDF)

**S1 Fig. Intermediate results in the image analysis workflow.**
(PDF)

**S2 Fig. Packaging assessment on AAV9 particles by CryoTEM.**
(PDF)

**S1 Appendix. Determination of the theoretical values of the percentage of full particles (% F) in the specimen used for the linearity assessment.**
(PDF)

**S2 Appendix. In silico study for the determination of a statistically relevant number of detected particles.** A graphical representation of the standard deviation for each sample size and theoretical full ratio.
(PDF)

## Acknowledgments

We gratefully acknowledge Ngarita Kingi for her help in proof-reading this manuscript.

## Author Contributions

**Conceptualization:** Mathieu Colomb-Delsuc, Roman Raim, Christian Fiedler, Stefan Reuberger, Johannes Lengler, Rickard Nordström, Martin Ryner, Ioana Mihaela Folea, Barbara Kraus, Juan A. Hernandez Bort, Ida-Maria Sintorn.

**Data curation:** Mathieu Colomb-Delsuc, Roman Raim, Christian Fiedler, Stefan Reuberger, Johannes Lengler, Rickard Nordström, Martin Ryner, Ioana Mihaela Folea.

**Formal analysis:** Mathieu Colomb-Delsuc, Roman Raim, Christian Fiedler, Stefan Reuberger, Johannes Lengler, Rickard Nordström, Martin Ryner, Ioana Mihaela Folea.

**Investigation:** Mathieu Colomb-Delsuc, Roman Raim, Christian Fiedler, Stefan Reuberger, Rickard Nordström, Ioana Mihaela Folea, Ida-Maria Sintorn.

**Methodology:** Mathieu Colomb-Delsuc, Roman Raim, Stefan Reuberger, Johannes Lengler, Martin Ryner, Ioana Mihaela Folea, Juan A. Hernandez Bort.

**Project administration:** Mathieu Colomb-Delsuc, Roman Raim, Stefan Reuberger, Barbara Kraus, Juan A. Hernandez Bort, Ida-Maria Sintorn.

**Resources:** Mathieu Colomb-Delsuc, Roman Raim, Rickard Nordström, Barbara Kraus, Juan A. Hernandez Bort, Ida-Maria Sintorn.

**Software:** Martin Ryner.

**Supervision:** Mathieu Colomb-Delsuc, Roman Raim, Johannes Lengler, Juan A. Hernandez Bort, Ida-Maria Sintorn.

**Validation:** Mathieu Colomb-Delsuc, Roman Raim, Rickard Nordström, Ioana Mihaela Folea, Barbara Kraus, Juan A. Hernandez Bort, Ida-Maria Sintorn.

**Visualization:** Mathieu Colomb-Delsuc, Roman Raim, Juan A. Hernandez Bort.

**Writing – original draft:** Mathieu Colomb-Delsuc, Roman Raim, Christian Fiedler, Stefan Reuberger, Johannes Lengler, Rickard Nordström, Martin Ryner, Ioana Mihaela Folea, Juan A. Hernandez Bort, Ida-Maria Sintorn.

**Writing – review & editing:** Mathieu Colomb-Delsuc, Roman Raim, Christian Fiedler, Stefan Reuberger, Johannes Lengler, Rickard Nordström, Martin Ryner, Barbara Kraus, Juan A. Hernandez Bort, Ida-Maria Sintorn.

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
