## [Decision Letter · Decision Letter 0]

21 Feb 2022

PONE-D-21-40321Assessment of the Percentage of Full Recombinant Adeno-Associated Virus Particles in a Gene Therapy Drug Using CryoTEMPLOS ONE

Dear Dr. Colomb-Delsuc,

Thank you for submitting your manuscript to PLOS ONE. After careful consideration, we feel that it has merit but does not fully meet PLOS ONE’s publication criteria as it currently stands. Therefore, we invite you to submit a revised version of the manuscript that addresses the points raised during the review process.

We look forward to receiving your revised manuscript.

Kind regards,

Chen Ling, Ph.D.

Academic Editor

PLOS ONE

Journal Requirements:

Reviewers' comments:

Reviewer's Responses to Questions

**Comments to the Author**

1. Is the manuscript technically sound, and do the data support the conclusions?

Reviewer #1: Yes

Reviewer #2: Partly

2. Has the statistical analysis been performed appropriately and rigorously? 

Reviewer #1: Yes

Reviewer #2: No

3. Have the authors made all data underlying the findings in their manuscript fully available?

Reviewer #1: Yes

Reviewer #2: No

4. Is the manuscript presented in an intelligible fashion and written in standard English?

Reviewer #1: Yes

Reviewer #2: No

5. Review Comments to the Author

Reviewer #1: In this manuscript, titled “Assessment of the Percentage of Full Recombinant Adeno-Associated Virus Particles in a Gene Therapy Drug Using CryoTEM”, Colomb-Delsuc and coworkers describe an analytical workflow of quantifying E/F ratios in various rAAV preparations using a cryogenic TEM workflow. The authors highlight the purported advantages of cryoTEM over other more commonly employed methods of quantifying E/F ratios, with a particular focus placed on negative stain TEM (nsTEM). The analytical method proposed by the authors would certainly be interesting to their target audience and thus deserving of publication.

However - while the manuscript contains a substantial quantity of numerical results, the subsequent analysis and interpretation of these data in some places (particularly when comparing cryoTEM results to other, non-nsTEM methods) seem somewhat superficial. The manuscript, as well as potential readers, would thus greatly benefit from a more detailed interpretation and analysis of the presented results.

Specific points:

1) One important aspect of any analysis method that is not discussed in the manuscript is the speed of analysis (i.e. throughput) vs. accuracy/resolution. Compared to other methods (e.g. chromatography or even nsTEM), how fast is the proposed cryoTEM method? How much sample is required compared to other analytical methods?

2) The comparison of cryoTEM to other analytical methods for E/F determination is presented in vague, superficial terms, such that it is difficult for the reader to assess the relative strengths/weaknesses of each approach (e.g. Page 4).

Which specific methods allow discrimination of sub-populations? Moreover, other analytical methods are also unaffected by serotype changes, transgene identity, etc. and could be explicitly stated. A noteworthy example of such a technique that is not at all mentioned in the manuscript are mass measurement-based approaches (e.g. charge detection mass spectrometry with recent examples PMID: 34631929 and PMID: 34977271). Would be good to discuss that as well.

3) Figure 2: The linear curve fit to the in vitro biopotency data in panel 2A is unusual in that it has a surprisingly large, non-zero y-intercept value (0.15), implying that even rAAV preparations containing no genome are biologically active. Can the authors comment on this unusual behaviour? As the figure uses E/F ratios obtained via cryoTEM, is it possible that there may be inaccuracies/biases in this method versus, for example AUC or vg/cp? It would be highly informative if the authors explicitly compared their biopotency data with E/F ratios for each sample derived from AUC or vg/cp, as already listed in Table 1, with a direct comparison with respect to e.g. linearity of the correlation.

4) Figure 3: The correlation between cryoTEM and vg/cp titre is presented here in a confusing manner. As both these methods fundamentally determine the same property (the proportion of filled and empty capsids), the x- and y-axes should be normalized to the same scaling (e.g. 0 – 100%, or 0.0 – 1.0). In an ideal case, the slope of this plot would then be exactly 1 (i.e. a perfect correspondence between cryoTEM and vg/cp). As is evident from the presented results, this is not the case, and could be further elaborated.

5) The authors include a nicely detailed discussion as to biases inherent to nsTEM that may complicate E/F quantitation using that technique. However, an analogous, equally important discussion for the other tested methods (e.g. AUC, vg/cp) is not present. While it is appreciated that the authors state that these discrepancies have already been previously observed in literature “without further insight”, the rAAV community would greatly benefit from a similarly thorough analysis as was presented for nsTEM.

Reviewer #2: This manuscript describes a validated method, based on cryo-electron microscopy, to quantify the composition of recombinant adeno associated virus preparations. This work is an important contribution to the increasingly important field of gene therapy approaches using viral vectors, and could also be applied to other viral systems without significant modification. The work describes a reliable approach to distinguish empty from full particles, by counting single molecules and arriving at an acceptable standard deviation of ~0.6 from repeat experiments with multiple grids. An important advantage of this approach is derived from the additional electron density imparted by the DNA cargo of filled particles. The ability to be able to clearly and visually distinguish filled from empty particles makes this approach very convincing and compelling. The statistical treatment of the cryo-TEM results is sound an credible. Alternative methods have well documented drawbacks. The discussion focused on discrepancies between cryo-TEM and nsTEM explains well potential reasons for the shortcomings of nsTEM.

As expected, comparison of cryo-TEM with other methods falls short, both in quantitation and in statistical consistency, except the correlation with in-vivo potency, which was a good test to include. I agree with the authors that the reasons for consistent comparisons are related to the well documented lack of reliability associated with the other methods chosen for comparison (qPCR, ELISA assays, etc). However, the gold standard characterization, when done correctly, is AUC, and it should provide a close to 100% correlation with the cryo-TEM results, at least when partially filled particles, contaminants, and aggregates are excluded from the comparison, which the authors admit they cannot do reliably with their method.

While the authors claim that their results show a good correlation between AUC and cryo-TEM, the results in Table 1 clearly show that this is actually far from fact. It is clear to see from Table 1 that there are big differences between cryo-TEM and AUC, especially when the percentages of full capsids are large. While the trends agree very nicely, the absolute quantification is far off when the relative amount of filled capsids increases. The reason for this discrepancy is never mentioned or discussed, although there is an obvious answer, and it is rooted in the flawed analysis of the AUC data. The correlation could be much better, and I believe, provide close to 100% correlation with the amount of filled capsids.

I feel the manuscript would be significantly strengthened if the AUC analysis would have been done correctly. The big flaw here is that the amount of capsids measured by AUC is based on a UV absorbance at 280 nm. The incorrect assumption here is that the concentration of filled, empty and partially filled particles corresponds simply to their extinction at 280 nm. This is clearly not the case, since the *molar* extinction coefficient of each type of particle varies as a function of the amount of loaded DNA cargo, because there is a significant overlap between the DNA and protein absorbance spectra at 280 nm. DNA and protein both absorb strongly at 280 nm (about 50% of the peak absorbance of DNA at ~260 nm), and the molar extinction coefficient of a single particle increases dramatically as the amount of incorporated DNA increases, since the capsid (protein) absorbance of 1 particle of course stays constant.

It therefore goes to reason that the absorbance based measurement is unreliable as presented, and since molar extinction corrections are not made, in fact, pretty much impossible in my opinion, this type of analysis cannot be used to accurately compare to the *NUMBERS* of particles (as counted in cryo-TEM) versus the absorbance of particles (as measured by 280 nm absorbance as done in the AUC). The authors are confusing a concentration measurement based on absorbance with a concentration measurement based on molar quantities. Since the cryo-TEM method *counts* particles, it is clear that a *molar* comparison must be used.

To fix this manuscript, the correct approach needs to take the differential absorbance of protein and DNA into account by measuring a multi-wavelength AUC experiment and deconvoluting the protein contribution, because ONLY the protein absorbance reflects the true capsid concentration, and those results must be compared to the cryo-TEM results. This was not done here, so the analysis compares apples to oranges. Not surprisingly, the percentage of filled capsids is highly exaggerated in the AUC analysis, and the discrepancy is increasingly distorted with an increase in the relative amount of filled capsids. For example, a 79% count of filled capsids for sample S1.10 compares to 96% filled capsids as measured by AUC (a 17% discrepancy), while the ratio of 2% filled capsids for sample S1.1 matches nicely with the AUC determined 2% when the amount of filled capsid is negligible. This means that AUC peaks corresponding to the filled capsids are exaggerating the amount of filled capsids present without adjustment for molar extinction. Since the molar extinction coefficients of various capsid species vary by serotype, DNA cargo loading, and single- vs. double stranded DNA conformation, molar extinction coefficients are notoriously difficult to obtain (some attempts have been made in this publication, using the same equipment as was used in this study:

https://pubmed.ncbi.nlm.nih.gov/34186069/

The approach demonstrated in this manuscript would be much more suitable for the comparison between AUC and cryo-TEM.

Minor points:

1. Why is the ICH Q2 validation procedure not discussed? It would be helpful to get more details on this.

2. The data availability statement was not provided. I cannot review primary data for correctness in this review, or access any data presented in this manuscript.

3. The use of a fixed frictional ratio of 1.4 for the analysis of all possible species in the mixture (empty, partial, filled) is incorrect for obvious reasons: particles containing different ratios of protein and DNA also have significantly different partial specific volumes, because the density of DNA and protein is very different. This leads to an overall variable partial specific volume, which quite dramatically affects the apparent frictional ratio when the partial specific volume is held constant, as it was done here. The authors should correct this by using a 2-dimensional approach like the C(s,f/f0) method to allow the frictional ratio to vary. While the overall shape of the particle clearly doesn't change as a function of DNA cargo loading, the frictional ratio, a theoretical construct, depends on an accurate partial specific volume value, which a) is not known here, and b) varies for each species. The easiest solution is to allow the frictional ratio to float.

4. The manuscript would benefit from improvements in the use of the English language (grammar, word choices, sentence structure). I recommend a revision by a native English speaker.

The authors have several choices to fix the manuscript:

1. repeat the AUC analysis using multi-wavelength analysis with appropriate 2-dimensional hydrodynamic treatment of the data

2. remove the flawed AUC data completely and stick to qPCR and other methods for validation. The cryo-TEM approach can stand on its own and trying to validate it with an improper analysis of AUC data severely detracts from the otherwise high quality of this manuscript, and perpetuates false data as being acceptable in publications. A flawed analysis should not be published, no matter how many other manuscripts make the same mistake.

If (1) is not an option, my second preference would be for (2) because the cryo-TEM work is beautifully performed with good linearity in the dilution series, solid statistics, and can stand perfectly on its own without the flawed AUC comparison. I do question the claimed validation according to the ICH Q2 (R1) guidelines without the proper orthogonal comparison to a gold standard AUC analysis, if a multi-wavelength analysis had been used. I realize that multi-wavelength analysis with the XLA is tedious, but for a validated method like cryo-TEM a correctly performed AUC analysis would be much preferred over a method like qPCR or ELISA. A true multi-wavelength analysis would require a repeat of the AUC experiments, and most likely the TEM experiments since the samples are probably no longer available. I therefore would be OK with (2). I would not agree to keeping the current AUC analysis (even if all flaws and mistakes were commented), because it perpetuates a serious problem with AUC analysis that is widespread throughout literature and is apparently tolerated with all its flaws by industry. The cryo-TEM approach represents a major improvement, such that short of a correctly performed AUC analysis, I would prefer it any day over any other method presented in this manuscript.

6. PLOS authors have the option to publish the peer review history of their article (what does this mean?). If published, this will include your full peer review and any attached files.

Reviewer #1: **Yes: **Albert J R Heck

Reviewer #2: No

---

## [Author Response · Author response to Decision Letter 0]

6 May 2022

Thank you for the comments and suggestion on our work. We are now please to submit a revised version of the manuscript in which the comments from the reviewers and editors have been addressed. Please see updated version of the cover letter and response to reviewer documents for more details on the changes and updates.

---

## [Editor Report · Decision Letter 1]

16 May 2022

Assessment of the percentage of full recombinant Adeno-Associated Virus particles in a gene therapy drug using CryoTEM

PONE-D-21-40321R1

Dear Dr. Colomb-Delsuc,

We’re pleased to inform you that your manuscript has been judged scientifically suitable for publication and will be formally accepted for publication once it meets all outstanding technical requirements.

Kind regards,

Chen Ling, Ph.D.

Academic Editor

PLOS ONE
---

## [Editor Report · Acceptance letter]

23 May 2022

PONE-D-21-40321R1 

Assessment of the percentage of full recombinant Adeno-Associated Virus particles in a gene therapy drug using CryoTEM 

Dear Dr. Colomb-Delsuc:

I'm pleased to inform you that your manuscript has been deemed suitable for publication in PLOS ONE. Congratulations! Your manuscript is now with our production department. 

Kind regards, 

on behalf of

Dr. Chen Ling 

Academic Editor

PLOS ONE